# Antibody escape and global spread of SARS-CoV-2 lineage A.27

Tamara Kaleta[1,22], Lisa Kern[1,22], Samuel Leandro Hong [2,22], Martin Hölzer[3], Georg Kochs [1], Julius Beer[1], Daniel Schnepf [1], Martin Schwemmle [1], Nena Bollen[2], Philipp Kolb [1], Magdalena Huber[1], Svenja Ulferts [4], Sebastian Weigang [1], Gytis Dudas [5], Alice Wittig [3,6], Lena Jaki[1], Abdou Padane[7], Adamou Lagare[8], Mounerou Salou[9], Egon Anderson Ozer [10], Ndodo Nnaemeka[11], John Kofi Odoom[12], Robert Rutayisire[13,14], Alia Benkahla[15], Chantal Akoua-Koffi[16], Abdoul-Salam Ouedraogo[17], Etienne Simon-Lorière [18], Vincent Enouf[19], Stefan Kröger [20], Sébastien Calvignac-Spencer [21], Guy Baele [2], Marcus Panning [1✉] & Jonas Fuchs [1✉]

In spring 2021, an increasing number of infections was observed caused by the hitherto rarely described SARS-CoV-2 variant A.27 in south-west Germany. From December 2020 to June 2021 this lineage has been detected in 31 countries. Phylogeographic analyses of A.27 sequences obtained from national and international databases reveal a global spread of this lineage through multiple introductions from its inferred origin in Western Africa. Variant A.27 is characterized by a mutational pattern in the spike gene that includes the L18F, L452R and N501Y spike amino acid substitutions found in various variants of concern but lacks the globally dominant D614G. Neutralization assays demonstrate an escape of A.27 from convalescent and vaccine-elicited antibody-mediated immunity. Moreover, the therapeutic monoclonal antibody Bamlanivimab and partially the REGN-COV2 cocktail fail to block infection by A.27. Our data emphasize the need for continued global monitoring of novel lineages because of the independent evolution of new escape mutations.

[1] Institute of Virology, Freiburg University Medical Center, Faculty of Medicine, University of Freiburg, Freiburg, Germany. [2] Department of Microbiology, Immunology and Transplantation, Rega Institute, KU Leuven, Herestraat 49, 3000 Leuven, Belgium. [3] Methodology and Research Infrastructure, Bioinformatics, Robert Koch Institute, Berlin, Germany. [4] Institute of Experimental and Clinical Pharmacology and Toxicology, Freiburg University Medical Center, Faculty of Medicine, University of Freiburg, Freiburg, Germany. [5] Gothenburg Global Biodiversity Centre, Carl Skottsbergs gata 22B, 413 19 Gothenburg, Sweden. [6] Hasso Plattner Institute, Digital Engineering Faculty, University of Potsdam, 14482 Potsdam, Germany. [7] Institut de Recherche en Santé, de Surveillance Epidémiologique et de Formation (IRESSEF), Rufisque, Senegal. [8] Centre de Recherche Médicale et Sanitaire (CERMES), 634 Bld de la Nation, BP:10887YN034 Niamey, Niger. [9] Laboratoire de Biologie Moléculaire et d'Immunologie, Département des Sciences Fondamentales, Université de Lomé, Lomé, Togo. [10] Center for Pathogen Genomics and Microbial Evolution, Institute for Global Health, Northwestern University Feinberg School of Medicine, Chicago, IL 60611, USA. [11] National Reference Laboratory, Nigeria Centre for Disease Control, Abuja, Nigeria. [12] Department of Virology, Noguchi Memorial Institute for Medical Research, University of Ghana, Legon, Accra, Ghana. [13] Rwanda National Joint Task Force COVID-19, Rwanda Biomedical Centre, Ministry of Health, Kigali, Rwanda. [14] National Reference Laboratory, Rwanda Biomedical Center, Kigali, Rwanda. [15] Laboratory of BioInformatics bioMathematics, and bioStatistics (BIMS), Institut Pasteur de Tunis, University of Tunis El Manar, Tunis, Tunisia. [16] Laboratoire Central, Centre Hospitalier et Universitaire de Bouaké, Bouaké, Côte d'Ivoire. [17] Department of Bacteriology and Virology, Souro Sanou University Hospital, Bobo-Dioulasso, Burkina Faso. [18] G5 Evolutionary Genomics of RNA Viruses, Department of Virology, Institut Pasteur, Paris, France. [19] National Reference Center for Respiratory Viruses, Institut Pasteur, Paris, France. [20] Department of Infectious Disease Epidemiology, Robert Koch Institute, 13353 Berlin, Germany. [21] Epidemiology of Highly Pathogenic Microorganisms, Robert Koch Institute, 13353 Berlin, Germany. [22]These authors contributed equally: Tamara Kaleta, Lisa Kern, Samuel Leandro Hong. ✉email: marcus.panning@uniklinik-freiburg.de; jonas.fuchs@uniklinik-freiburg.de

The continuing pandemic spread of SARS-CoV-2, the causative agent of coronavirus disease 2019 (COVID-19), has a devastating global impact on life, health care systems and economies by causing significant morbidity and mortality in the human population. SARS-CoV-2 is an enveloped, positive-sense single-stranded RNA virus and infects host cells via binding of the viral spike glycoprotein (S) to the angiotensin-converting enzyme 2 (ACE2) receptor and proteolytic activation through cellular proteases[1,2]. The mature S protein is cleaved into two subunits S$_1$ and S$_2$ and organized as a homotrimer in the viral particle[3]. While S$_1$ forms a globular structure essential for ACE2 binding, S$_2$ mediates membrane fusion. Both the receptor-binding domain (RBD) and the N-terminal domain (NTD)[4] are targeted by neutralizing antibodies in sera of convalescent and vaccinated individuals[5,6]. Thus, multiple RBD-specific monoclonal antibodies (mAb) are assessed in clinical trials or are approved to treat COVID-19, including Bamlanivimab (LY-Cov-555) in combination with Etesevimab (LY-COV016)[7] and the REGN-COV2 mAb cocktail (REGN10933 and REGN10987)[8].

Early in the pandemic, SARS-CoV-2 acquired the S D614G substitution that has been associated with increased transmissibility and set the genetic foundation for the large number of B.1 derived lineages[9,10]. As the pandemic progressed the genomic diversity of SARS-CoV-2 increased significantly and several variants of concern (VOCs) and variants of interest (VOIs) emerged. These variants may be associated with higher transmissibility, can lead to more severe disease and/or significantly escape from antibody-mediated immunity, thereby reducing the effectiveness of available vaccines and treatments with mAbs[11–13]. Prominent examples are the B.1.1.7 (Alpha) and B.1.617.2 (Delta) variants that dominated global infections in late 2020 and 2021. These variants are characterized by specific patterns of concerning S mutations: apart from the D614G substitution, lineage B.1.1.7 has the N501Y amino acid substitution associated with increased affinity to ACE2[14,15] and two deletions in the NTD, among other changes. Moreover, a B.1.1.7 sub-lineage with an additional E484K substitution in the RBD has been detected in multiple countries. The E484K amino acid change is also found in other VOCs/VOIs and has been shown to reduce antibody neutralization[16]. A prominent amino acid change in the S protein of the B.1.617.2 variant is L452R that is also found in various other lineages. This mutation was shown to enhance infectivity in vitro and decrease neutralization by sera of COVID-19 patients and vaccinees[17,18].

Here, we describe the detection, inferred origin and phenotypic characteristics of SARS-CoV-2 lineage A.27, which was primarily identified in Germany[19] and France[20] in spring 2021. This variant emerged in late 2020 and spread to over 30 countries. Through travel history-aware phylogeographic reconstruction, we estimate Western Africa as the likely origin of this lineage, from where it spread to other regions. A.27 is characterized by a mutational profile including L18F, L452R and N501Y in the S protein that combines genetic changes found in various VOCs/VOIs, while lacking the D614G substitution present in most other lineages. Our data demonstrate that A.27 can partially escape the neutralization by sera of vaccinees and recovered COVID-19 patients, and by therapeutic mAbs. This study emphasizes the importance of continued molecular surveillance to quickly detect antibody escape variants that threaten global vaccination efforts.

## Results

**Detection of SARS-CoV-2 lineage A.27.** In early 2021, sequencing of material from SARS-CoV-2 infected individuals in Germany increased with about one third of sequences generated in the south-western state Baden-Wuerttemberg (BW) located at the French border (Fig. 1a/b). Within the molecular surveillance program of the Robert Koch Institute (RKI, Public Health Institute Germany), 851/178.264 (0.48%) sequences were classified as lineage A.27 from 18th of January to 1st of June 2021. The A.27 lineage was initially defined in January 2021 following an outbreak in Mayotte[21], an overseas region of France located in the Indian ocean between the coast of Mozambique and Madagascar.

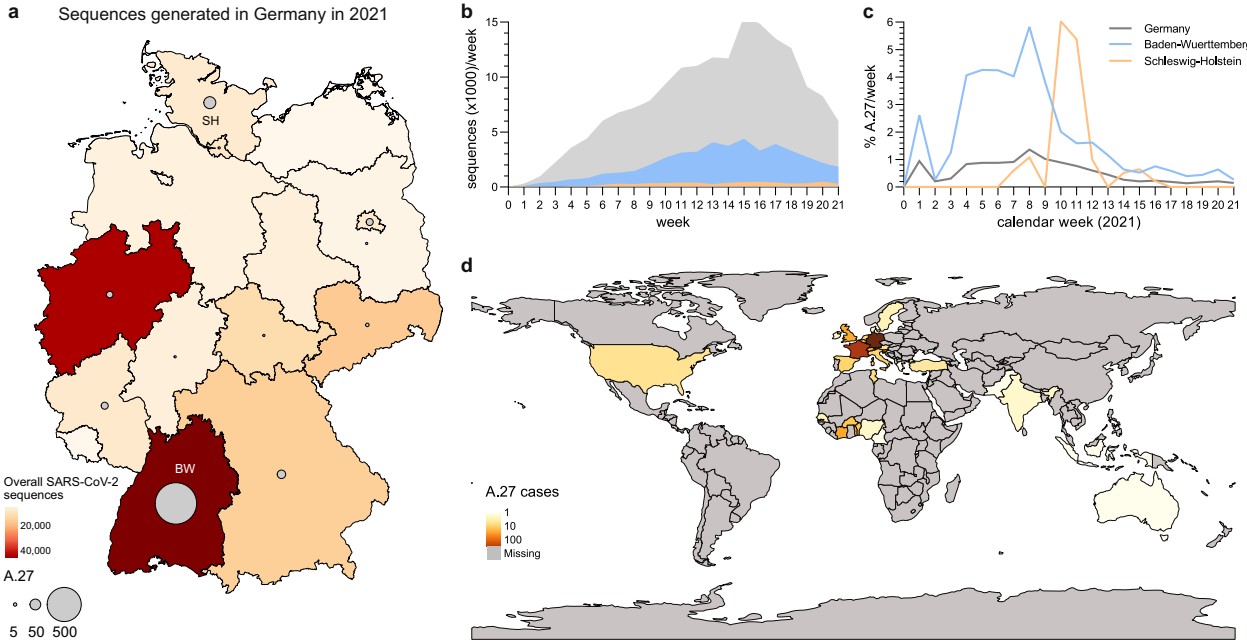

**Fig. 1 Detection of lineage A.27 in Germany. a** Map of Germany displaying the cumulative number of all SARS-CoV-2 sequences (colors) and A.27 sequences (circles) generated in different federal states between January and June 2021. BW—Baden-Wuerttemberg, SH—Schleswig-Holstein. **b**, **c** Temporal distribution of sequences for each calendar week for (**b**) all or (**c**) A.27 sequences in Germany and the two federal states BW and SH. **d** World map displaying the cumulative A.27 sequences obtained from the GISAID and RKI databases. Source data are provided as a Source Data file.

Most of the German A.27 sequences originated in BW (81.4%), and in the northern state of Schleswig-Holstein (SH) (6.8%) (Fig. 1a). In the beginning of 2021, the frequency of A.27 cases steadily increased, reaching up to 6% of all sequences generated in BW and SH (Fig. 1c). Afterwards, the relative detection rate of this variant decreased while the frequency of VOC B.1.1.7 consistently increased (Supplementary Fig. 1). From December 2020 to June 2021, A.27 was reported in 31 countries. Apart from the German sequences, 535 A.27 sequences had been deposited in the GISAID database[22]. The earliest sequences were reported in Denmark (Europe) in December 2020 and the Western African countries Senegal, Burkina Faso and Togo. Interestingly, the majority of A.27 sequences deposited in GISAID between January and April 2021 originated from France ($n = 263$) indicating a similarly rapid spread in France compared to Germany (Fig. 1d).

**Phylogeographic analyses reveal West Africa as the likely origin of A.27.** To further characterize the global spread of A.27 and to estimate the potential origin of this lineage, we performed maximum-likelihood (ML) phylogenetic and Bayesian phylogeographic analyses incorporating available travel data of 12 A.27 infected patients (Supplementary Table 1). A preliminary unrooted ML phylogenetic analysis of 1386 complete A.27 genomes and a representative set of 2516 sequences from an Africa-focused Nextstrain build, capturing the global SARS-CoV-2 diversity, showed sufficient temporal signal in a root-to-tip regression analysis. Therefore, we estimated a global time-calibrated phylogeny, which dated its most recent common ancestor (tMRCA) to the second half of November 2020. The predicted evolutionary rate of $7.60e^{-04}$ substitutions per site and year was in line with previous estimates for SARS-CoV-2[23]. We subsequently performed a more targeted ML phylogenetic analysis on a subtree of the inferred global phylogeny that contained all A.27 and 25 non-A.27 genomes. The analysis of this subset also had a sufficient temporal signal and a time calibrated tree showed that A.27 is monophyletic and further diversified in early 2021 (Supplementary Fig. 2). Location-specific clusters within Germany or France indicated possible independent introductions of A.27 into Europe.

Subsampled travel history-aware Bayesian phylogeographic analyses of the A.27 subtree (Supplementary Fig. 2) using BEAST 1.10.5[24] estimated the ancestral origin of the A.27 lineage in Western Africa (Fig. 2a and Supplementary Fig. 3). The subsampling was performed to take the sampling bias toward the high proportion of sequences from Germany and France into account (Fig. 2b). This analysis predicted a tMRCA for A.27 in late September 2020 (95% Highest Posterior Density interval (HPD) ranging between mid-August 2020 and late October 2020) with an evolutionary rate of $8.15e^{-04}$ (95% HPD: [$7.04e^{-04}$; $9.33e^{-04}$]) substitutions per site and year. The earliest introduction of A.27 into Germany likely occurred in the third week of November 2020 (95% HPD covering the second half of November 2020), while the introduction into France happened slightly earlier around the beginning of November 2020 (95% HPD between early October 2020 and mid-November 2020). This places the estimated tMRCAs a few months before the first confirmed cases in both countries (January 4th and January 6th of 2021 for France and Germany, respectively). Therefore, we estimate that A.27 was introduced into Europe between 6 and 8 weeks after its common tMRCA.

We estimate that A.27 was initially transmitted within Western Africa (Fig. 2a and Supplementary Fig. 3) before spreading to other regions. The spread from West Africa was estimated by the expected number of transitions between all regions (Markov jumps). We confirmed a large number of introductions out of

West Africa and multiple separate introduction events in Germany and France, leading to the different large German and French clades (Fig. 2c and Supplementary Fig. 4). Other African regions could have been the source for seeding introductions into Europe, such as Mayotte seeding into France, although this was not consistently supported (Supplementary Data 1). However, we also inferred introductions from France to the Benelux Union and Northern Africa, as well as introductions into Asia-Pacific (APAC) from the Benelux Union. Although no A.27 sequences were available to us from Southern Africa, our travel history-aware phylogeographic analysis showed consistent and strong support for a spread to this region from Western Europe, which in turn led to an introduction into regions of the Benelux Union. This bi-directional exchange of lineages between African regions and the European continent can also be observed for Northern and Central Africa. This is in contrast with Western and Eastern Africa, which were seen as exclusive sources of lineages in Europe in our reconstruction. In conclusion, A.27 likely originated in Western Africa in late 2020, from where it spread to multiple countries around the globe resulting in large clusters in Germany and France in spring 2021.

**Proportion of hospitalized patients infected with A.27 or B.1.1.7 in Germany.** As part of the German molecular surveillance program, sequences uploaded to the RKI are linked to case-based data reported by local public health authorities into the electronic surveillance system for infectious disease. We compared available patient data of 329 sequenced A.27 and 56,453 patient specimens of VOC B.1.1.7 between January and June of 2021. Interestingly, in the set of sequences that were randomly selected and limited to not fully vaccinated patients, those infected with A.27 ($n = 100$) were on average 5.6 years older than B.1.1.7 ($n = 17.512$) infected individuals (Fig. 3a). There was no gender difference in terms of A.27 patients compared to B.1.1.7 patients (Fig. 3b). Accordingly, we compared the proportion of hospitalized B.1.1.7 and A.27 infected patients. We found no significant differences in the proportion of hospitalization, 8.9% for B.1.1.7 and 6.2% for A.27, respectively (Fig. 3c). Analysis of A.27 and B.1.1.7 infections shows, that A.27 infection also preferentially occur in older individuals with an increasing risk for hospital admissions by age.

**A.27 is characterized by a mutation profile similar to current VOCs and VOIs.** We characterized the mutational profile of A.27 based on 1386 full genome A.27 sequences. The nucleotide profiles were determined in comparison to Wuhan-Hu-1 using covSonar (https://gitlab.com/s.fuchs/covsonar) and aligned to each other (Supplementary Fig. 5). Lineage-specific mutations were defined as mutations present in ≥75% of all sequences (Table 1). The A.27 lineage is characterized by 26 mutations including seven non-synonymous mutations in the S gene, a frameshift in ORF3a and a deletion in ORF8. The frameshift in ORF3a is located at the C-terminus in a region that has so far not been resolved in available cryo-electron microscopy analyses[25] and leads to a 14 amino-acid truncated protein. The deletion in ORF8 also resides at the C-terminal end and translates into a deletion of an aspartate and phenylalanine involved in the stabilization of the ORF8 dimerization interface[26] (Supplementary Fig. 6). Three of the seven S substitutions, L18F, L452R, and N501Y, are of particular interest (Fig. 4a–c). The L18F amino acid substitution is found within the first of five loops of the NTD supersite[5,27] and the L452R and N501Y mutations are located in the receptor-binding motif (RBM) which interacts with the human ACE2 protein[1,4,28]. Both regions are targets for neutralizing antibodies[27,29,30] and L18F and L452R have been

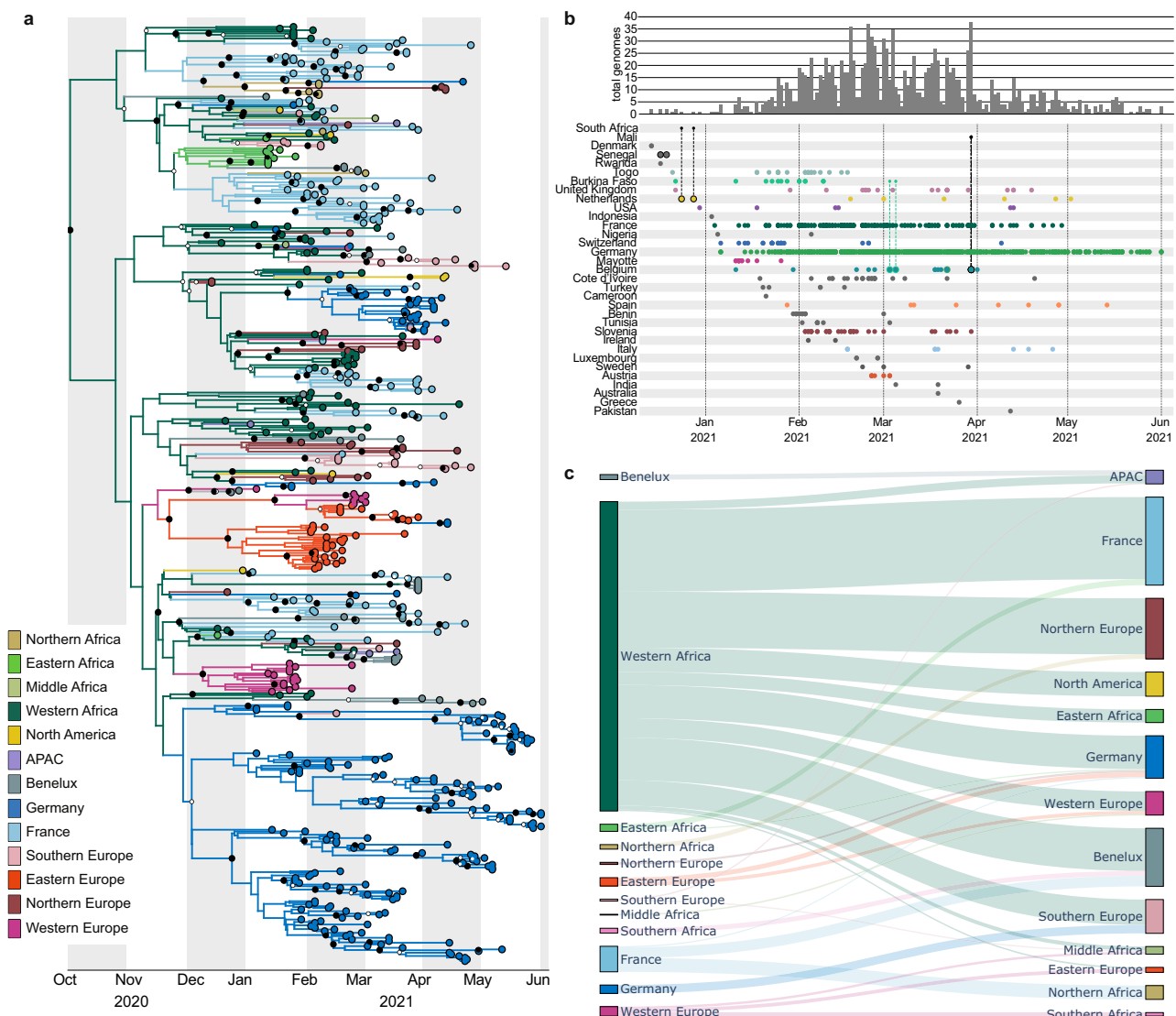

**Fig. 2 Origins of SARS-CoV-2 lineage A.27. a** Region-annotated phylogeny, showing an inferred origin of the A.27 lineage in Western Africa with subsequent spread to most of the other regions where A.27 was detected. Smaller white circles represent posterior support >0.5, whereas bigger black circles represent posterior support >0.95. **b** Known locations, dates and individual travel histories of A.27 cases. Rows show the collection dates of genomes on the bottom, as well as the frequency of genomes as a bar plot at the top. The origin and destination is shown for travel cases: genomes with associated travel history are outlined with the color corresponding to the origin location, and are connected to this origin location with a smaller dot. **c** Sankey plot showing the number of transitions between locations through the estimated number of Markov jumps. The thickness of the lines are proportional to the number of Markov jumps from the location to the left into the location on the right, conditional on the corresponding Bayes factor being higher than 3, pointing to strong support for an inferred origin of the A.27 lineage in Western Africa. Input XML files of the phylogeographic analysis are supplied in the Supplementary Data 1.

previously associated with antibody escape and L452R additionally with increased infectivity[17,18,31]. Furthermore, N501Y was suggested to enhance the binding affinity to ACE2[14,15]. These three mutations are also found in multiple VOCs and VOIs (Fig. 4c). L18F is present in the VOCs B.1.351 and P.1, the L452R substitution is found in high frequencies in B.1.617.2 and related AY lineages, and N501Y is known from B.1.1.7, B.1.351 and P.1. One of the hallmarks of A.27 is the absence of the S D614G substitution present in the globally dominating B.1-derieved lineages, indicating an independent acquisition of the other spike mutations.

**The A.27 Black Forest isolate is attenuated in vivo.** To characterize the biological features of the A.27 lineage, we isolated this variant from an oropharyngeal swab. The sample was derived

from a patient living in the Black Forest area in South Germany and who was treated at the University Medical Center of Freiburg. Virus isolation was performed on VeroE6 cells and followed by one cell culture passage to produce high titre stocks. We performed whole genome sequencing of the initial patient material and the derived virus stocks to analyze if the virus had acquired cell culture adaptations during isolation. Analysis of the variant frequencies found in the respective samples showed a ~60% variant frequency for the G11083T substitution in the ORF1ab after isolation. Although likely selected during isolation, this mutation was already present in low frequencies in the patient material (Fig.5a). Moreover, the virus isolate exhibited high genomic stability throughout the isolation process and all lineage-defining mutations were confirmed. The A.27 Black Forest isolate reached similar titres in both VeroE6 and Calu3

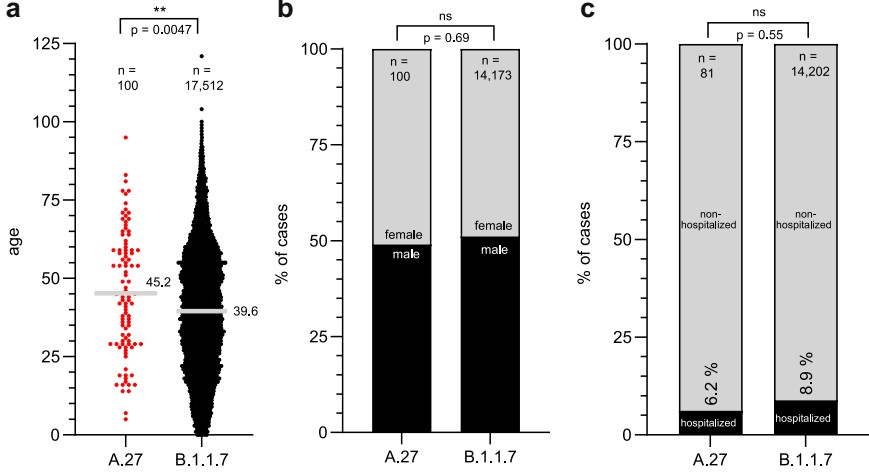

**Fig. 3 Metadata analysis of patients infected with A.27.** Metadata of confirmed A.27 and B.1.1.7 infected patients was acquired from the RKI and GISAID. **a** Scatter plot of the age distribution of patients infected with B.1.1.7 or A.27 patients. Displayed is the mean age. Statistical analysis was performed with a two-sided *t*-test (\*\**p* < 0.01). **b** Gender distribution and (**c**) hospitalization rate of B.1.1.7 and A.27 patients. Statistics were performed with a two-sided Fisher's exact test (ns— not significant). Source data are provided as a Source Data file.

**Table 1 Consensus mutations of lineage A.27 (n = 1386).**

| 75% of sequences | 90% of sequences | gene | amino acid substitution |
|---|---|---|---|
| A361G | A361G | ORF1a | synonymous |
| C1122T | C1122T | ORF1a | P286L |
| C2509T | C2509T | ORF1a | synonymous |
| C8782T | C8782T | ORF1a | synonymous |
| A9204G | A9204G | ORF1a | D2980G |
| A11217G | A11217G | ORF1a | N3651S |
| C16293T | – | ORF1b | synonymous |
| C16466T | C16466T | ORF1b | P5401L |
| A18366G | A18366G | ORF1b | synonymous |
| A20262G | A20262G | ORF1b | synonymous |
| C21614T | C21614T | S | L18F |
| G22468T | G22468T | S | synonymous |
| T22917G | T22917G | S | L452R |
| A23063T | A23063T | S | N501Y |
| C23520T | – | S | A653V |
| C23525T | C23525T | S | H655Y |
| G23948T | G23948T | S | D796Y |
| G25218T | G25218T | S | G1219V |
| T25541C | T25541C | ORF3a | V50A |
| del:26160:8 | – | ORF3a | del257/258fsX6 |
| C27247T | C27247T | ORF6 | synonymous |
| T28144C | T28144C | ORF8 | L84S |
| del:28248:6 | del:28248:6 | ORF8 | del:119/120 |
| A28273T | A28273T | NCR | synonymous |
| G28878A | G28878A | N | S202N |
| G29742A | – | NCR | synonymous |

cells, comparable with a prototypic B.1 isolate (Muc-IMB-1)[32] that only harbors the S D614G substitution in its viral genome and four different VOCs (Fig. 5b/c). A clear exception was the B.1.351 variant, which showed a 100-fold reduced viral titre 3 days post infection compared with A.27 on Calu3 cells (Fig. 5c) as previously reported[33]. Furthermore, cells infected with the B.1 and A.27 isolates were analyzed by confocal immunofluorescence microscopy (Fig. 5d). Both virus isolates showed a diffuse cytosolic accumulation of the S and N proteins 8 h post infection. The frameshift in ORF3a in the A.27 isolate might result in an altered cellular localization of this viral protein. Therefore, we additionally stained for ORF3a, but found no differences between B.1 and

A.27 infected cells indicating that the missing C-terminal amino acids do not impact its localization. This is in line with previous results showing a comparable cellular localization of wild type ORF3a and a C-terminal deletion mutant of ORF3a[34]. A previous study showed that ORF3a represents an important virulence factor in human ACE2 transgenic (hACE2) mice[35]. Therefore, we hypothesized that the ORF3a frameshift in the A.27 isolate could lead to an attenuated phenotype in vivo. To test this hypothesis, we compared the pathogenicity of B.1, B.1.1.7 and A.27 isolates in hACE2 transgenic mice[36]. Mice were infected with an intranasal inoculum containing 132 plaque forming units (pfu) per virus variant and weight loss and survival were monitored. B.1 and B.1.1.7 infected mice showed pronounced weight loss and all mice reached humane endpoints between 6 and 7 days post infection (Fig. 5e/f). Intriguingly, 75% of A.27 infected mice only transiently lost weight and recovered from the infection demonstrating that A.27 is severely attenuated in vivo compared to the B.1 and B.1.1.7 isolates, possibly due to the deletion in ORF3a.

**A.27 escaped neutralization by patient sera and therapeutic antibodies.** Lineage A.27 has two mutations in its RBD that translate to L452R and N501Y. Previous binding studies of RBD mutants with mAb and sera showed decreased binding for both mutations[6,31] (Supplementary Fig. 7a/b). To estimate the effect of the mutations found in the S gene of A.27 in the context of virus neutralization, serial dilutions of sera from convalescent COVID-19 patients (Fig.6a and Supplementary Fig. 8) or BioNTech BNT162b2 vaccinees (Fig. 6b and Supplementary Fig. 9) were analyzed by plaque reduction assays. The potential escape was assessed by comparing the A.27 Black Forest isolate to the prototypic B.1 isolate that harbors the D614G mutation. The neutralizing titres resulting in 50% plaque reduction ($NT_{50}$) of sera from convalescent COVID-19 patients and from vaccinees were significantly reduced, on average, two- to three-fold lower against the A.27 isolate compared to the B.1 isolate. Notably, the resistance of A.27 toward antibody neutralization was similar in either group as there were no significant differences in the $NT_{50}$ values between convalescent and vaccinee sera (Fig. 6c). Furthermore, we assessed the escape of A.27, B.1 and four different VOC isolates from the neutralizing capacity of the mAbs LY-COV555[37] and the REGN-COV2 cocktail (REGN10933, REGN10987)[8] which can be used to treat COVID-19 patients. In strong contrast to B.1 and B.1.1.7, the A.27 as well as the B.1.351, B.1.617.2 and

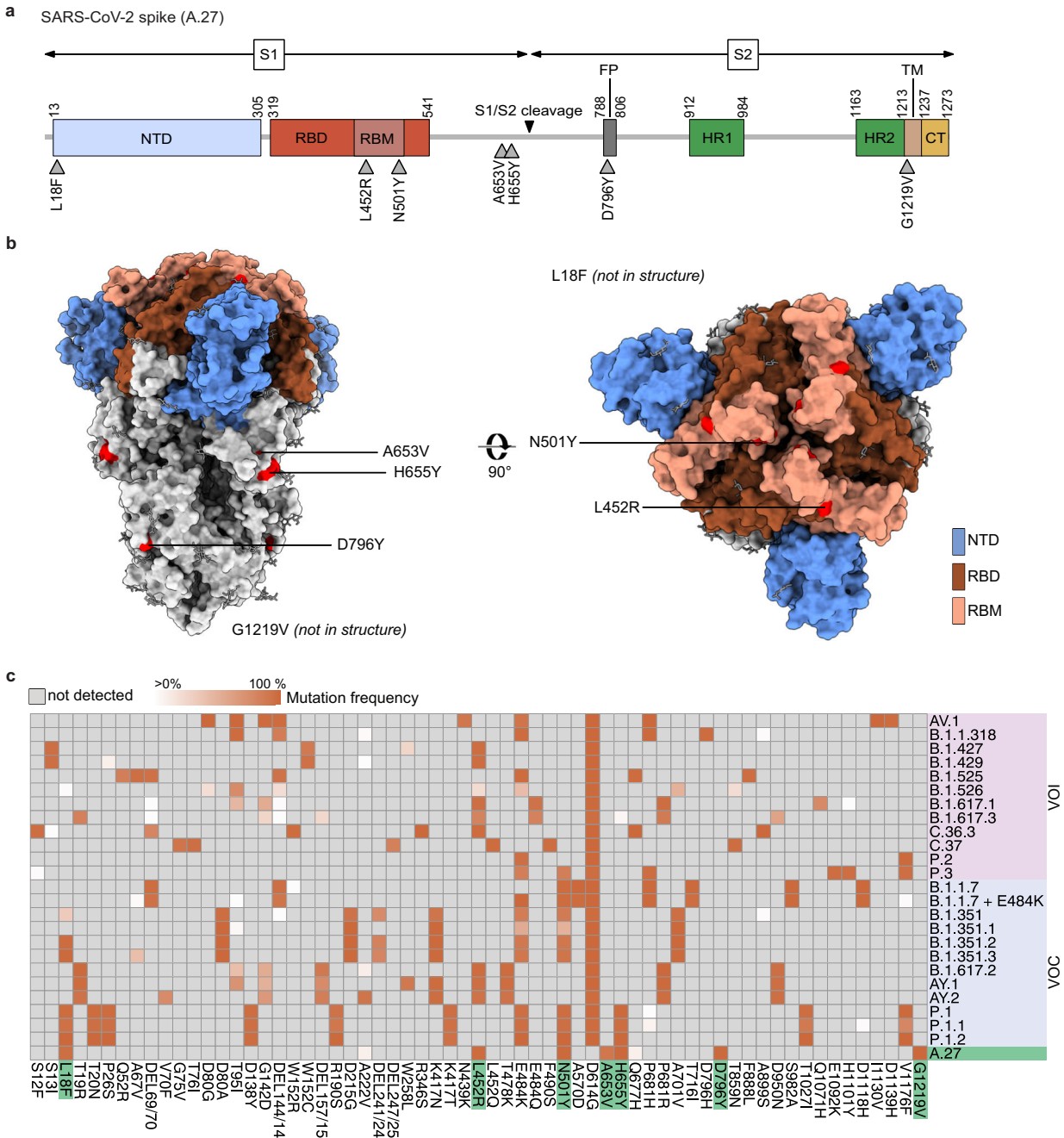

**Fig. 4 A.27 shows a mutational pattern in its viral spike gene that resembles other VOCs and VOIs. a** Schematic overview of the SARS-CoV-2 spike gene and the non-synonymous mutations found in more than 75% of A.27 sequences ($n = 1386$) compared to the S of Wuhan-Hu-1 sequence (NC_045512.2). NTD–N-terminal domain, RBD–receptor-binding domain, RBM–receptor-binding motif, FP–fusion peptide, HR1/2–heptad repeat region, TM–transmembrane domain, CT–C-terminal domain. **b** The spike protein of SARS-CoV-2 (PDB accession number: 6vxx) with the mutations displayed in (**a**) is shown in surface presentation. **c** Comparison of spike mutations present in A.27 and the different VOCs and VOIs. The mutation frequencies of the different lineages were downloaded from outbreak.info[70] (accessed on 2021-07-13) and the A.27 mutation frequencies replaced with the frequencies calculated based on the 1386 sequences used in this study. The heatmap was visualized with the R pheatmap package. Source data are provided as a Source Data file.

P.1 isolates completely escaped the neutralizing effect of LY-COV555 (Fig. 6d). For REGN10933, B.1.351 and P.1 displayed a pronounced escape (Fig. 6e). Furthermore, the neutralizing capacity of REGN10987 against A.27 and B.1.617.2 was slightly reduced (Fig. 6f). The observed differences for the individual REGN-COV2 antibodies could be compensated by a 1:1 combination of both antibodies, mimicking the actual treatment regimen[8] (Fig. 6g). Based on the $NT_{50}$ values, REGN10987 showed

an overall broad and strong neutralizing capacity while LY-COV555 failed to neutralize most variants (Table 2). Besides neutralization, a prime function of antigen-bound (complexed) IgG is the activation of Fc-gamma receptors (FcγRs) present on various immune cells such as monocyte-derived cells and Natural Killer (NK) cells. One of the most potent antiviral immune mechanisms mediated by the Fc-part of complexed IgG (Fcγ) is antibody-dependent cellular cytotoxicity elicited by NK cells

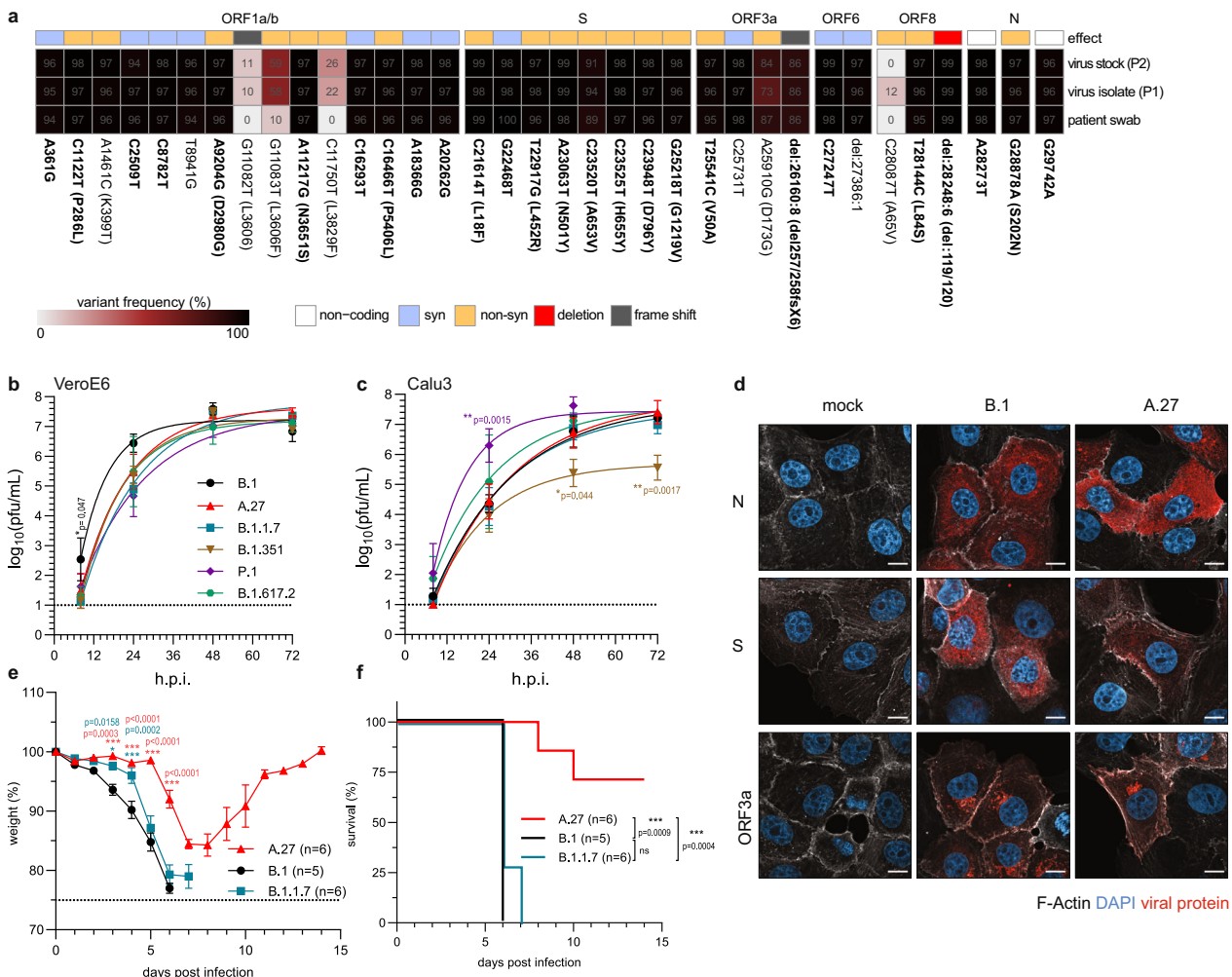

**Fig. 5 A.27 Black Forest isolate displays all lineage-defining mutations and is attenuated in vivo. a** Variant frequency plot (https://github.com/jonas-fuchs/SARS-CoV-2-analyses) of the variant frequencies detected by next-generation sequencing in the oropharyngeal swab of a patient from the Black Forest area in Germany infected with A.27, after virus isolation (passage 1/P1) and virus cultivation (passage 2/P2) on VeroE6 cells. The variant frequencies in comparison to Wuhan-Hu-1 (NC_045512.2) are plotted as a heatmap with the respective frequencies indicated as %. Mutations present in 75% of A.27 sequences are marked bold. **b**, **c** Viral growth kinetics of the A.27/Freiburg isolate (P2) in comparison to the virus isolates B.1, B.1.1.7, B.1.351, P.1 and B.1.617.2 in (**b**) VeroE6 or (**c**) Calu3 cells. Confluent cells were infected (moi of 0.001) and the cell supernatant was harvested after 8, 24, 48 and 72 h post infection. Viral titres were determined by plaque assay on VeroE6 cells. The log-transformed titres are shown as means ± SD from three independent experiments. Significance was determined in comparison to A.27 via a two-way ANOVA with a Tukey´s multiple comparison test, *$p < 0.05$, **$p < 0.01$. **d** Confocal fluorescence microscopy analysis of B.1 or A.27 infected VeroE6 cells (moi 0.1) 8 h post infection. Fixed cells were stained with SARS-CoV-2 N, S and ORF3a specific antibodies (red). In addition, F-actin (phalloidin, white) and nuclear DNA (DAPI, blue) were detected. Shown are representative pictures of two independent experiments. Scale bars indicate 10 μm. **e**, **f** Weight loss (**e**) and survival (**f**) of hACE2 transgenic mice infected intranasally with 132 pfu of the A.27 ($n = 7$), B.1 ($n = 5$) or B.1.1.7 ($n = 7$) isolates were monitored daily (mean ± SEM). Significance in weight loss was determined in comparison to B.1 via a two-way ANOVA with a Tukey´s multiple comparison test, *$p < 0.05$, ***$p < 0.001$. Significance for the survival was calculated with a Log-rank (Mantel–Cox) test (ns—not significant, ***$p < 0.001$). Source data are provided as a Source Data file.

expressing FcγRIII/CD16. Therefore, we assessed the potential of the above mAbs to activate CD16 in a cell-based FcγR-activation reporter assay. Inactivated virions from different strains were titrated and immobilized on ELISA plates and incubated with the respective mAbs at a fixed concentration. CD16 reporter cells were then cultured on opsonized virions and IL-2 production was measured as an indicator of receptor activation as described previously[38]. Directly immobilized mAbs served as a positive control and showed that all mAbs are equally able to activate CD16 (Fig. 6h). Incubation on opsonized A.27 and B.1.617.2 virions resulted in reduced CD16 activation for LY-COV555 in line with the neutralization data (Fig. 6i). However, for the other isolates there was no direct correlation between CD16 activation

and neutralizing capacity. Considering that all mAbs were able to activate CD16, this showed that residual binding of the therapeutic antibodies to the antigen could still result in CD16 activation. Taken together, these data argue for a pronounced escape of A.27 from neutralizing antibodies similar to other VOCs.

## Discussion

Here, we report the epidemiology, inferred origin and phenotypic characteristics of SARS-CoV-2 lineage A.27, whose genome contains several substitutions that have also been observed in various VOCs and VOIs. The root of the pandemic lies in the parental A lineage with the characteristic C8782T and T28144C

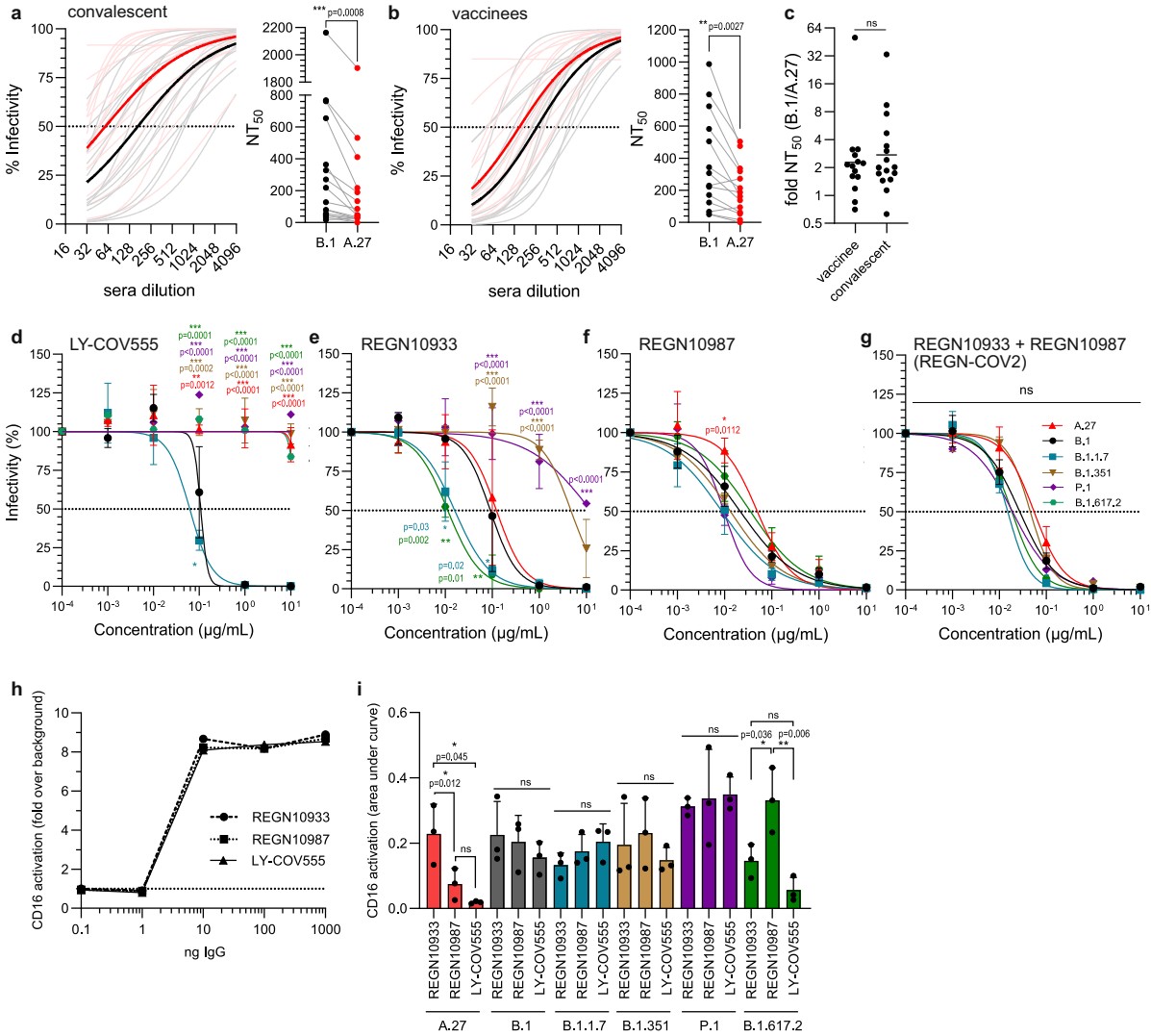

**Fig. 6 Neutralizing capacity of sera and therapeutic monoclonal antibodies against A.27. a** Neutralizing activity of convalescent sera ($n = 16$) or (**b**) sera from BioNTech BNT162b2 vaccinees ($n = 14$) against B.1 (gray) or A.27 (red). 100 pfu of each virus was incubated with serial twofold sera dilutions and analyzed by plaque assay. The curve fits (light color) and the mean curve fit (dark color) were plotted (left panel) and neutralization titres calculated (right panel). Statistics were performed with a paired, two-tailed $t$-test (**$p < 0.01$, ***$p < 0.001$). **c** The fold difference of the $NT_{50}$ between B.1 and A.27 was calculated for the analyzed convalescent sera ($n = 16$) and sera from vaccinees ($n = 14$). Shown are the individual values and the geometric mean. Statistics were performed on log2-transformed values with a two-tailed $t$-test (ns—not significant). **d–g** Neutralizing capacity of the therapeutic antibodies (**d**) LY-COV555, (**e**) REGN10933, (**f**) REGN10987 or (**g**) the combination of REGN10933 and REGN10987 in a 1:1 ratio. Serial tenfold dilutions of the monoclonal antibodies were incubated with 100 pfu of the different viruses and analyzed by plaque assay. Plotted are the curve fits and mean ± SD of three independent experiments. Statistics were calculated with a two-way ANOVA (Tukey's multiple comparison test, *$p < 0.05$, **$p < 0.01$, ***$p < 0.001$). **h, i** CD16 reporter cell IL-2 production was quantified via anti-mIL-2 ELISA. REGN10987, REGN10933 or LY-COV555 were (**h**) titrated and immobilized or (**i**) inactivated virions of different SARS-CoV-2 isolates were titrated, immobilized and opsonized with 20 ng/μl of IgG. In (**e**) the mean of two independent experiments is displayed. The dotted horizontal line in (**h**) represents the ELISA background set to 1. For (**i**) the area under the curve was calculated from virion titration (undiluted, 1:10,1:100). Shown are mean and standard deviation of three independent experiments. Statistics were performed with a one-way ANOVA (Tukey's multiple comparison test, ns—not significant, *$p < 0.05$, **$p < 0.01$). Source data are provided as a Source Data file.

mutations. However, lineages derived from A are rare, with only a few thousand sequences reported worldwide. In the COVID-19 pandemic, infections are currently dominated by B.1-derived lineages harboring the prominent D614G mutation[9]. This makes the mutational pattern of lineage A.27 of particular interest as it shows the independent acquisition of the same mutations found in B.1-derived VOCs and VOIs in a different genomic background. A similar pattern of concerning mutations in an emerging A-derived lineage has so far only been described for the A.23/A.23.1 lineage first discovered in Uganda and Rwanda[39,40].

The early detection in multiple countries in Western Africa in late December 2020, an outbreak in January 2021 in Mayotte and A.27 infections of Belgian military personnel returning from Mali pointed to a suspected origin in the African continent[41,42]. Through carefully crafted phylogeographic analyses that exploit individual travel histories of patients infected with A.27, we here provide support for the origin of A.27 in Western Africa. After completion of our phylogenetic and phylogeographic analyses, on the 17th of September 2021, three additional A.27 genomes from Burkina Faso appeared on GISAID with sampling dates on the 16

**Table 2 NT$_{50}$ values of the different monoclonal antibodies against the tested variants.**

|  | LY-COV555 | REGN10933 | REGN10987 | REGN-COV2 |
|---|---|---|---|---|
| A.27 | >10 µg/ml | 0.12 µg/ml | 0.05 µg/ml | 0.05 µg/ml |
| B.1 | 0.11 µg/ml | 0.09 µg/ml | 0.02 µg/ml | 0.03 µg/ml |
| B.1.1.7 | 0.06 µg/ml | 0.02 µg/ml | 0.01 µg/ml | 0.02 µg/ml |
| B.1.351 | >10 µg/ml | 4.70 µg/ml | 0.01 µg/ml | 0.05 µg/ml |
| P.1 | >10 µg/ml | >10 µg/ml | 0.01 µg/ml | 0.02 µg/ml |
| B.1.617.2 | >10 µg/ml | 0.01 µg/ml | 0.04 µg/ml | 0.02 µg/ml |

and the 19th of December 2020. They additionally confirm the early circulation and our inferred origin of A.27 in West Africa toward the end of 2020. The entire backbone of the A.27 phylogeny was estimated to be located in Western Africa, with the virus consequently spreading from there to other, mostly European, regions. From Western Africa, we observed that A.27 spread to Europe through multiple separate introduction events, eventually forming several large German and French clades. Interestingly, the Markov jump analysis did not support bi-directional seeding events between the two adjacent countries, which was further supported by the clear phylogenetic separation of the French and German clades. Note that due to the effect of sampling bias, for example as a result of varying sequencing efforts between countries, the major German and French clades do not necessarily signify that the spread of A.27 was largely confined to these two countries. This was confirmed by the individual travel histories we collected, with two infected patients testing positive in the Netherlands after traveling back from South Africa, although South Africa did not yet report any A.27 genomes.

While the origin of most French A.27 clades also lay in Western Africa, one of the larger French clades seemed to be more closely related to Eastern-African sequences, a fact also corroborated by the Markov jump analysis. This Eastern-African clade corresponds to an outbreak in Mayotte, a French overseas territory in the Indian Ocean, indicating the possibility that one or more travel cases from Mayotte to France led to the introduction of A.27 into France. Notably, only one out of five replicates (see Supplementary Data 1) of our phylogeographic analysis provides strong support (Bayes factor > 20) for A.27 being introduced into France via Mayotte, with the other four replicates showing positive support (Bayes factor > 3). Known travel cases from Mayotte to France of patients infected with A.27 could have led to more conclusive evidence, but unfortunately, we were not able to obtain such individual travel histories for the purpose of our travel history-aware phylogeographic analyses. Of note, we were also not able to obtain travel history for the earliest A.27 sample in Denmark which we assume to be a travel case given that all of the other early A.27 samples came from West Africa.

Within Germany, particularly in BW, we observed a constant increase in A.27 sequences over several weeks followed by a rapid decline of A.27. Since the increase of B.1.1.7 started earlier, this decline occurred with the start of the third wave in Germany, which was mainly driven by B.1.1.7 and led to its dominance. This suggests a fitness advantage of A.27 in comparison to other lineages but a disadvantage compared to the B.1.1.7 lineage. A possible explanation for this intermediate fitness phenotype could be the acquisition of the N501Y mutation in the absence of D614G. Both mutations are present in VOCs B.1.1.7, B.1.351 and P.1 and might have additive effects. N501Y increases the affinity of the viral S for ACE2[14] and D614G is thought to prevent premature dissociation of the S trimer leading to a higher infectivity and transmissibility[10,43,44]. We detected a lower, but not a significant lower proportion of hospitalization of A.27 compared to

B.1.1.7 cases, but a significant mean higher age. However, these observations were clearly limited by the small amount of available metadata. To investigate if both viruses had a comparable virulence, we analyzed the growth of the A.27 isolate in cell culture as well as its pathogenicity in hACE2 transgenic mice. The A.27 and B.1.1.7 isolates demonstrated a comparable viral replication in cell culture but A.27 was significantly attenuated in mice compared to B.1 and B.1.1.7. This indicates that despite comparable viral replication in cell culture, A.27 possesses features which decrease its pathogenicity in mice. Genomic changes like the frameshift in ORF3a and the deletion in ORF8 might contribute to this phenotype. The lack of position 119/120 in the accessory protein ORF8 might lead to a decreased stability of the homodimer and a partial loss of function[26]. Interestingly, B.1.617.2 lacks the two ORF8 amino acids 120/21, strongly indicating a convergent evolution for deletions in this region or the lack of a selective pressure to maintain these amino acids. ORF8 seems to be dispensable for viral replication and deletions/stop codons could represent an adaptation to the human host[45]. ORF3a has been shown to induce apoptosis and block autophagy[34,46]. Our immunofluorescence analysis suggests a similar cellular localization and expression of ORF3a in B.1 and A.27 infected cells. However, the frameshift in ORF3a might impair some of the proteins' functions. An attenuation due to a crippled ORF3a would be in line with a previous study showing that ORF3a and ORF6 are the major contributors of viral pathogenesis in hACE2 transgenic mice[35]. Implications of these amino acid changes for human disease are presently unclear. Functional characterizations of mutations in accessory viral proteins of SARS-CoV-2 are urgently needed to better understand their impact on virulence and pathogenicity of SARS-CoV-2 in humans.

Vaccinations are currently the major instrument to combat the COVID-19 pandemic and treatments with mAb are a potent therapeutic option to treat COVID-19. As such, antibody escape mutations in circulating variants could further fuel the pandemic. A.27 harbors multiple mutations in the viral S, the major target for neutralizing antibodies, raising the question whether mAb and sera from COVID-19 patients or from vaccinees will protect from an A.27 infection. Our data suggest that A.27 can escape antibody-mediated immunity. We observed a consistent decrease of the neutralizing capacity of sera from COVID-19 patients and vaccinees. Moreover, A.27 completely escaped the neutralization of LY-COV555 and partially of REGN10987. Importantly, B.1.617.2 escaped these antibodies in a similar manner, suggesting that the L452R mutation present in both variants facilitates this escape[47]. This is in agreement with deep mutational scanning data that showed decreased binding of L452R and LY-COV555[31]. Interestingly, the VOCs B.1.351 and P.1 escaped LY-COV555 and REGN10933 which was likely facilitated by the E484K mutation present in both variants. This suggests that L452R and E484K lead to an escape from LY-COV555[31] and to a partial resistance to either REGN10987 or REGN10933, respectively. The FcγR activation assay showed that this escape is to some extend independent of the CD16 activation. However, reduced neutralization can result in reduced CD16 activation as observed for LY-COV555 and A.27 or B.1.617.2 indicating key immunological mechanisms linked to opsonisation can be additionally impaired in non-neutralizing mAbs. Importantly, the combination of both REGN-COV2 antibodies sufficiently neutralized A.27 as well as all tested VOCs, emphasizing that REGN-COV2 but not LY-COV555 should be used to treat COVID-19 patients suffering from an infection with these variants. This also argues for using different mAb preparations when no clear clinical effect is observed, as the lineage might harbor mutations that are refractory to a particular preparation. Notably, NTD polymorphisms might also decrease the neutralizing capacity of sera as multiple

studies have detected NTD-specific antibodies in COVID-19 patients[48] and vaccinees[49]. L18F lies within the first of five loops of the NTD supersite[5,27] and could contribute to the escape from antibody-mediated immunity as previously suggested[50].

The present study analyzed the genomic profile and biological features of the A.27 lineage which was primarily detected in France and Germany in spring 2021. Our phylogeographic analyses that were able to exploit individual travel histories provided evidence that these sequences stem from separate introduction events out of Western Africa, which we estimate to be the geographic origin of A.27. Importantly, our data further suggest that A.27 is less susceptible to SARS-CoV-2-specific antibodies and that COVID-19 patients and vaccinees might not be fully protected against this variant. The presence of concerning S mutations in an A-derived lineage supports the notion that the same escape mutations can appear in relatively distant genomic backgrounds with similar phenotypic consequences. Therefore, global molecular surveillance has to continue to detect novel variants and to support assessing their risk for the human population.

## Methods

**Data acquisition.** To assess the prevalence of lineage A.27 in Germany, all SARS-CoV-2 full-genome sequences that were submitted to the RKI ($n = 851$) in 2021 until the 1st of June (Supplementary Data 2) were classified using the PANGO lineage assignment (pangolin version: 3.0.3, pangoLEARN: 2021-05-27)[51,52]. The relative frequency of A.27 and B.1.1.7 sequences in comparison to all submitted sequences was assessed for each federal state and each week of 2021. Sequencing data was linked to patient metadata obtained from local health authorities as part of the genomic surveillance program of the RKI via the national electronic reporting system for surveillance of notifiable infectious diseases (SurvNet)[53]. Anonymized data of the hospitalization status, sex and age were extracted. Furthermore, as of 2021-08-24 all additionally available 535 A.27 sequences and associated metadata for samples taken until 2021-06-01 were downloaded from GISAID (Supplementary Table 2). The Germany map was downloaded from https://gadm.org/maps/DEU.html and the maps were visualized using tmap[54]. Patient metadata of hospitalization status, sex and age was analyzed with GraphPad Prism.

### Phylogenetic analysis and time-calibrated phylogenetic tree reconstruction.
To put the A.27 sequences within the global context, we combined all 851 A.27 sequences from the RKI with 535 A.27 sequences from GISAID, along with 2545 sequences from the Africa-focused Nextstrain build (https://nextstrain.org/ncov/gisaid/africa)[55] for a total of 3907 sequences after removing duplicate entries. We limited our selection of sequences from GISAID and Nextstrain to those collected up to June 1st to remain consistent with the sampling period of the RKI sequences. We used this selection of sequences to infer an unrooted phylogenetic tree using IQTREE2 v2.1.0[56] under a GTR model with empirical frequencies and four-category FreeRate model of site heterogeneity, which was selected as the best fitting model using IQTREE's ModelTest functionality. The resulting phylogeny was then time calibrated using TreeTime v.0.7.4[57], rooting the tree on the "Wuhan/Hu-1/2019" isolate, following the Nextstrain SARS-CoV-2 workflow (https://github.com/nextstrain/ncov) and assuming a strict molecular clock and a skyline coalescent model. TreeTime detected one GISAID sequence (EPI_ISL_1586901) and three RKI sequences (IMS-10020-CVDP-DCAB86B5-00C8-496F-9B16-297546A77DF2, IMS-10122-CVDP-DF8FAF93-3173-40A6-85D8-B50274A72B20, IMS-10122-CVDP-5B313DE6-7B34-4BAA-81BD-DEEE937126EC) as outliers, which we subsequently removed from further downstream analyses. Such a relatively low number of outliers is to be expected as the Nextstrain workflow already performs a similar data cleaning step. We visualized the resulting phylogeny using baltic (https://github.com/evogytis/baltic).

### Travel history-aware phylogeographic reconstruction.
From the full time-calibrated ML phylogeny, we selected the subtree containing all A.27 sequences, along with 25 non-A27 ancestral sequences ($n = 1383$) to perform a more targeted reconstruction to determine the geographic origin of the A.27 lineage. To this end, we aimed to perform a Bayesian phylogeographic reconstruction using BEAST v1.10.5[58]. We note that over 50% of the sequences in the A.27 clade were collected in Germany ($n = 884$), with France a close second ($n = 263$) (see Fig. 2b and Supplementary Fig. 2). Such severe sampling bias is known to affect discrete phylogeographic reconstruction[59], leading to overconfidence in inferring oversampled locations as ancestral in the phylogeny[60]. To mitigate this, we employed a subsampling scheme where we removed identical sequences and limited our dataset to a maximum of 10 randomly selected sequences per week for these two

locations. We performed this subsampling procedure five times, to exclude the possibility of accidentally sampling a highly unlikely scenario. This yielded final datasets of between 560 and 565 sequences from 31 countries, on which we performed travel history-aware phylogeographic reconstruction[24]. However, estimating transition rates between locations that have very few sequences may be subject to poor mixing[61]. In order to avoid this issue, we aggregated certain locations into larger regions (with the categorization based mainly on the UN geoschemes). For example, sequences from Denmark, Sweden, UK and Ireland were grouped together as belonging to "Northern Europe". We refer to Supplementary Table 3 for a detailed description of which countries were grouped into which regions and how many sequences were included in total per region. This process resulted in a total of 14 regions being considered in the phylogeographic analysis: Asia-Pacific (APAC), Benelux (Belgium, Netherlands and Luxembourg), Eastern Africa, Middle Africa, Southern Africa, Western Africa, North America, Eastern Europe, Southern Europe, Western Europe, France and Germany. We decided not to group France and Germany into a larger region, given that they are countries of interest for this study. For twelve sequences in our dataset that were sequenced in the Netherlands and Belgium, we obtained travel information, indicating cases in which a patient had travelled in the days preceding diagnosis. Two patients had returned from South Africa to the Netherlands, two from Burkina Faso to Belgium (and one other of whom a household member returned from Burkina Faso to Belgium and tested positive) and seven from Mali to Belgium (Fig. 2a and Supplementary Table 1).

With these regions and individual travel histories in place, we performed travel history-aware discrete phylogeographic analysis[24,59] (using BEAST 1.10.5[58], while employing the BEAGLE 3.2.0 high-performance computational library[62] to improve performance. We made use of Bayesian stochastic search variable selection to simultaneously determine which migration rates are zero depending on the evidence in the data and to efficiently infer the ancestral locations, in addition to providing a Bayes factor test to identify significant non-zero migration rates. We also estimated the expected number of transitions (known as Markov jumps[63]) between all regions in the dataset. On the sequence data partition, we made use of a general time-reversible substitution model with estimated base frequencies and among-site rate heterogeneity[64], along with a relaxed molecular clock model with an underlying lognormal distribution[65]. We used the following prior specifications for these analyses: a non-parametric skygrid coalescent model (for which we employed Hamiltonian Monte Carlo estimation[66]), a gamma (shape = 0.001; scale = 1000) prior on the skygrid precision parameter, dirichlet ($\alpha 1, \ldots \alpha K = 1.0$; K equal to the number of states) priors on the transition rates for the GTR substitution model and the frequencies for the GTR nucleotide-substitution model, an exponential (mean = 0.5) prior on the shape parameter of the discretized gamma distribution to model among-site rate heterogeneity, a Poisson prior (mean = 13) on the sum of non-zero rates between regions, a CTMC reference prior on the mean evolutionary rate[67] and an exponential (mean = 1/3) prior on its standard deviation. For the travel history-aware phylogeographic model, we treated the departure time of the patient as a random variable, conditioned on a normal prior distribution with a mean of 10 days before sampling date (based on a mean incubation time of 5 days and a constant ascertainment period of 5 days between symptom onset and testing[68]) and a standard deviation of 3 days. We truncated the distribution to be positive (back-in-time), in order to avoid an infection time at a later date than the corresponding sampling time.

Each of these phylogeographic analysis replicates ran for a total of 560 million iterations, respectively, with the Markov chains being sampled every 50,000th iteration, in order to reach an effective sample size (ESS) for all relevant parameters of at least 200, as determined by Tracer 1.7[69]. We used TreeAnnotator to construct maximum clade credibility (MCC) trees for each replicate.

### Analysis of A.27 lineage-defining mutations and lineage comparison.
The nucleotide and amino acid profiles of the 1386 A.27 sequences were determined in comparison to Wuhan-Hu-1 using covSonar (https://gitlab.com/s.fuchs/covsonar). To extract the nucleotide mutations and define INDELs the R package stringr was utilized. The profiles were matched with the R package dplyr, mutations with a frequency below 1% were excluded and the resulting matrix visualized with the R pheatmap package. We extracted mutations that present in 75% of the mutation profiles and defined them as lineage-defining mutations. Furthermore, the data produced by covSonar was utilized to compare the A.27 amino acid profile in the viral spike gene with different VOCs and VOIs. Here, the amino acid profile was subset for the viral spike gene and the frequency of the mutations calculated excluding again frequencies below 1%. The amino acid mutation frequencies in the viral spike of A.27 and the different VOCs and VOIs was downloaded from outbreak.info[70] on 2021-07-13. Outbreak.info calculates these frequencies based on all available sequence data from GISAID. The A.27 frequencies were replaced with our calculated frequencies as they include data from GISAID and RKI and visualized the mutation frequencies in the viral spike again with the R pheatmap package.

### Visualization of viral protein structures.
The EM structure of the closed of the trimeric SARS-CoV-2 spike protein (10.2210/pdb6vxx/pdb) and the dimeric 2.04 Å crystal structure of ORF8 (10.2210/pdb7JTL/pdb) were downloaded from the protein data bank and visualized with UCSF ChimeraX version: 1.1 (2020-09-09).

**Cell culture**. Virus isolation, cell culture and mouse infection experiments with SARS-CoV-2 were performed under Biosafety Level 3 protocols at the Institute of Virology, Freiburg, approved by the Regierungspraesidium Tuebingen (No. 25-27/8973.10-18 and UNI.FRK.05.16-29). Adherent African green monkey kidney VeroE6 cells (ATCC CRL-1586™) and human lung Calu3 cells (ATCC HTB-55™) were cultured in 1× Dulbecco's modified Eagle medium (DMEM) containing 5% or 10% fetal calf serum (FCS), respectively. To isolate SARS-CoV-2 from patient material, filtered throat swab samples of patients with previous SARS-CoV-2 A.27 (EPI_ISL_3200835) or Delta variant B.1.617.2 infections (EPI_ISL_2535433) were inoculated on VeroE6 cells ($2 × 10^6$ cells) in 4 ml DMEM with 2% FCS and incubated at 37 °C and 5% $CO_2$ for 4–6 days until a cytopathic effect was visible. The culture supernatant was cleared and stored at −80 °C. Virus titres were determined by plaque assay on VeroE6 cells. Furthermore, the following SARS-CoV-2 isolates were used: Muc-IMB-1, lineage B.1 (EPI_ISL_406862 Germany/BavPat1/2020)[32], kindly provided by Roman Woelfel, Bundeswehr Institute of Microbiology; Alpha variant B.1.1.7 (EPI_ISL_751799) and Beta variant B.1.351 (hCoV-19/Germany/NW-RKI-I-0029/2020; ID: EPI_ISL_803957), provided by Donata Hoffmann and Martin Beer, Friedrich-Loeffler-Institute, Riems; and Gamma variant P.1 (EPI_ISL_3980444) provided by Michael Schindler, Institute for Medical Virology and Epidemiology, Tuebingen. All virus stocks used for experiments were inspected for mutations compared to the parental virus isolate using whole genome Illumina sequencing. For the analysis of viral growth, VeroE6 or Calu3 cells were inoculated in six well plates with a moi of 0.001 and supernatant collected at 8 h, 24 h, 48 h and 72 h post-infection. Viral titres were then determined by plaque assay on VeroE6 cells. BW5147 mouse thymoma cells (kindly provided by Ofer Mandelboim, Hadassah Hospital, Jerusalem, Israel) stably express human FcγR ectodomains genetically fused to the CD3ξ signalling module[38]. Cells were maintained at $3 × 10^5$ to $9 × 10^5$ cells/ml in Roswell Park Memorial Institute medium (RPMI GlutaMAX, Gibco) supplemented with 10% (vol/vol) FCS, sodium pyruvate (1×, Gibco), 100 U/ml penicillin-Streptomycin (Gibco) β-mercaptoethanol (0.1 mM, Gibco). Cells were cultured at 37 °C, 5% $CO_2$. All cell lines were routinely tested for mycoplasma.

**Evaluation of the neutralizing capacity of sera and monoclonal antibodies**. Serological neutralization tests were performed with patient beta collected after resolved infection with SARS-CoV-2 or sera of vaccinated individuals ~10–50 days post-vaccination with the second dose of the BNT162b2 mRNA vaccine (Pfizer/BioNTech). Neutralizing antibody titres were determined by a plaque reduction assay. Serial twofold dilutions of the sera were incubated for 1 h with 100 pfu of the SARS-CoV-2 isolates. The serum-virus mixture was then used to infect VeroE6 for 90 min at room temperature. The inoculum was removed and the cells overlaid with 0.6% oxoid agar for 48 h at 37 °C. Cells were fixed with 3.7% formaldehyde and stained with crystal violet. The reduction in counted plaque numbers was determined in comparison to an untreated mock-infected control without serum. Neutralization titres of mAb were determined by incubation of the respective SARS-CoV-2 isolates with tenfold dilutions of the individual antibodies ($10^1$–$10^{-4}$ μg/mL). Plaque reduction assay was performed as described above replacing sera dilutions with serial dilution of the mAbs from concentration $10^1$–$10^{-4}$ μg/ml. To evaluate the neutralizing capacity and determine the neutralizing titre 50, a non-linear fit least squares regression (constraints: 0 and 100) was performed. For sera, the mean of each dilution for all sera was determined and plotted to visualize the overall tendency. The fold difference was calculated by the quotient of the $NT_{50}$ for B.1 and A.27 for the individual sera.

**Fc receptor activation assay**. Virus stocks were concentrated by ultracentrifugation at 100.000 $g$ for 2 h and 4 °C and the pellet dissolved in PBS. The concentrated virus stocks were inactivated with 0.1% β-propiolactone for 16 h at 4 °C followed by 2 h at 37 °C. IgG or the inactivated virions were titrated in PBS and incubated on an ELISA plate for 1 h at 37 °C for coating. Plates were then blocked in PBS with 10% FCS for 1 h at RT. Immobilized virions were opsonized by incubation with 20 ng/μl of the respective mAbs (IgG) for 2 h at RT, followed by incubation with mouse BW5147 reporter cells stably expressing human FcR ectodomains genetically fused to CD for 16 h in an incubator (37 °C, 5% CO2). Immobilized IgG was incubated with reporter cells directly. Secreted mIL-2 was quantified via anti-mIL-2 sandwich ELISA as described previously[38,71].

**Whole genome sequencing**. cDNA was produced from extracted RNA of oro-pharyngeal swab or cell culture supernatant samples using random hexamer primers and Superscript III (ThermoFisher) followed by a PCR tiling the entire SARS-CoV-2 genome (ARTIC V3 primer sets; https://github.com/artic-network/artic-ncov2019). This produced ~400 bp long, overlapping amplicons that were subsequently used to prepare the sequencing library. The amplicons were purified with AMPure magnetic beads (Beckman Coulter). Afterwards the QIAseq FX DNA Library Kit (Qiagen) was used to prepare indexed paired-end libraries for Illumina sequencing. Normalized and pooled sequencing libraries were denatured with 0.2 N NaOH. These libraries were sequenced on an Illumina MiSeq using the 300-cycle MiSeq Reagent Kit v2.

The de-multiplexed raw reads were subjected to a custom Galaxy pipeline, which is based on bioinformatics pipelines on usegalaxy.eu[72]. The raw reads were pre-processed with fastp v.0.20.1[73] and mapped to the SARS-CoV-2 Wuhan-Hu-1 reference genome (Genbank: NC_045512) using BWA-MEM v.0.7.17[74]. For datasets produced with the ARTIC v3 protocol, primer sequences were trimmed with ivar trim v1.9 (https://andersen-lab.github.io/ivar/html/manualpage.html). Variants (SNPs and INDELs) were called with the ultrasensitive variant caller LoFreq v2.1.5[75], demanding a minimum base quality of 30 and a coverage of at least 20×. Afterwards, the called variants were filtered based on a minimum variant frequency of 10 % and on the support of strand bias. The effects of the mutations were automatically annotated in the vcf files with SnpEff v.4.3.1[76]. Finally, consensus sequences were constructed with bcftools v.1.1.0[77]. Regions with low coverage (>20×) or variant frequencies between 30 and 70% were masked with N. The final consensus sequences have been deposited in the GISAID database (www.gisaid.org).

The clades of the reconstructed viral genomes were classified with the Pangolin webserver (pangolin.cog-uk.io). An in-house R script was also used to plot the variant frequencies that were detected by LoFreq as a heatmap (github.com/jonas-fuchs/SARS-CoV-2-analyses). This tool is also available on usegalaxy.eu ("Variant Frequency Plot").

**Immunofluorescence analysis**. VeroE6 cells seeded on glass coverslips were either infected with SARS-CoV-2 isolates at a moi of 0.1 or left uninfected. At 8 h post infection, cells were fixed in 4% paraformaldehyde in PBS, permeabilized with 0.3% Triton X-100 and blocked in 10% FCS. SARS-CoV-2 N- (Rockland #200-401-A50, 1:1000), S- (Rockland #600-401-MS8, 1:250) and ORF3a-specific primary antibodies (https://mrcppu-covid.bio/, 1:100) and AF568-labeled goat-anti-rabbit (Invitrogen, #A11011, 1:400) secondary antibody as well as AF488-labeled Phalloidin (Hypermol, #8813-01, 1:400) were used for staining. The coverslips were embedded in Diamond Antifade Mountant with 4′,6-diamidino-2-phenylindole (DAPI) (ThermoFisher, #P36971). Fluorescence images were generated using a LSM800 confocal laser-scanning microscope (Zeiss) equipped with a 63X, 1.4 NA oil objective and Airyscan detector and processed with Zen blue software (Zeiss) and ImageJ/Fiji.

**Infection of K18-hACE2 transgenic mice**. Transgenic (K18-hACE2)2Prlmn mice[36] were purchased from The Jackson Laboratory and bred locally. Hemizygous 8–12-week-old males were used in accordance with the guidelines of the Federation for Laboratory Animal Science Associations and the National Animal Welfare Body. All experiments were in compliance with the German animal protection law and approved by the animal welfare committee of the Regierungspraesidium Freiburg (permit G-20/91). Mice were anesthetized using isoflurane and infected intranasally (i.n.) with virus dilutions in 40 μl PBS containing 0.1% BSA. Mice were monitored daily and euthanized if severe symptoms were observed or body weight loss exceeded 25% of the initial weight.

**Plotting and statistical analysis**. All plots and statistics were generated with GraphPad Prism v8.4.2 or R studio (R version 4.0.2).

**Ethical statement**. The project has been approved by the ethical committee of the Albert-Ludwigs-Universität, Freiburg, Germany. Written informed consent was obtained from all participants and the study was conducted according to federal guidelines, local ethics committee regulations (Albert-Ludwigs-Universität, Freiburg, Germany: No. F-2020-09-03-160428 and no. 322/20) and the Declaration of Helsinki (1975). All routine virological laboratory testing of patient specimens (virus isolation and next-generation sequencing) was performed in the Diagnostic Department of the Institute of Virology, University Medical Center, Freiburg (Local ethics committee no. 1001913). Convalescent sera and sera of vaccinees were obtained from the Hepatology-Gastroenterology-Biobank as part of the Freeze-Biobank Consortium at the University Medical Center Freiburg. Written informed consent was obtained from all blood donors prior to inclusion.

**Reporting summary**. Further information on research design is available in the Nature Research Reporting Summary linked to this article.

## Data availability

All necessary data and information are given in the paper. Source data are provided with this paper. Input XML files of the phylogeographic analysis is supplied in the Supplementary Data 1. The sequence data were submitted to the GISAID data base and are publicly available (Supplementary Table 2). Note, that due to sequencing or reconstruction errors (e.g., causing frameshifts) not all A.27 genome sequences obtained from external laboratories could be uploaded to GISAID. However, all sequences and metadata obtained from the RKI are also available via https://github.com/robert-koch-institut/SARS-CoV-2-Sequenzdaten_aus_Deutschland, including also all A.27 sequences used in this study (Supplementary Data 2). Raw sequencing data have been submitted to the European Nucleotide Archive (https://www.ebi.ac.uk/ena/browser) under the study accession number: ERP134884. Source data are provided with this paper.

## Code availability

The script to visualize the variant frequencies is publicly available (github.com/jonas-fuchs/SARS-CoV-2-analyses, v1.0) and implemented on usegalaxy.eu (Variant Frequency Plot).

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

## Acknowledgements

We thank Roman Woelfel (Bundeswehr Institute of Microbiology) for providing the B.1 (Muc-IMB-1) isolate; Donata Hoffmann and Martin Beer (Friedrich-Loeffler-Institut, Insel Riems) for providing the B.1.1.7 and B.1.351 isolates, Michael Schindler (Institute for Medical Virology and Epidemiology, Tuebingen) for providing the P.1 isolate; Markus Hoffmann (Goettingen) for the Calu-3 cells, Todd Giardiello (Rockland Immunochemicals PA) for providing anti-N and anti-S specific rabbit antisera and James Hastie (MRC Protein Phosphorylation and Ubiquitylation Unit, College of Life Sciences, University of Dundee) for providing polyclonal sheep antibody targeting ORF3a (see https://mrcppu-covid.bio/). We gratefully acknowledge the authors from the originating laboratories responsible for obtaining the specimens and the submitting laboratories where genetic sequence data were generated and shared via the DESH hub of the RKI and GISAID (Supplementary Data 2 and Supplementary table 3). We acknowledge the contribution of all local and state public health authorities, laboratories, and health workforce who have submitted COVID-19 case-based data to the German notification system. We would like to thank Bas B. Oude Munnink (Department of Viroscience, WHO Collaborating Centre for Arbovirus and Viral Hemorrhagic Fever Reference and Research, Rotterdam, Netherlands) for providing travel history information of A.27 cases. We are furthermore grateful for the sequencing efforts from our colleagues Judd F. Hultquist, Ramon Lorenzo-Redondo and Lacy M. Simons (Center for Pathogen Genomics and Microbial Evolution, Institute for Global Health, Northwestern University Feinberg School of Medicine, Chicago, IL 60611, USA), Olubusuyi M. Adewumi (Department of Virology, College of Medicine, University of Ibadan, Ibadan, Nigeria), Dr. Halatoko Wemboo INH (Institut National d'Hygiene, Lomé, Togo), Dagnra Anoumou Yaotsè (Biolim Université de Lomé, Lome, Togo), Ilhem Boutiba, Riadh Gouider, Sameh Trabelsi, and Henda Triki (Laboratory of BioInformatics, bioMathematics and bioStatistics (BIMS)) and Dr. Justin Lee (CDC Atlanta). We furthermore like to acknowledge the excellent technical assistance of Valentina Wagner and Annette Ohnemus. The authors are grateful to Zsolt Ruzsics, Walter Haas and Otto Haller for helpful comments on the paper. This work was funded by the Deutsche Forschungsgemeinschaft (DFG, German Research Foundation, grant number PA 2274/4-1), by the Bundesministerium fuer Bildung und Forschung (BMBF) through the Deutsches Zentrum fuer Luft- und Raumfahrt, Germany to M.P. and M.S (DLR, grant number 01KI2077) and to A.S.O. and C.A.K. (ANDEMIA; grant number 01KA1606). S.L.H. acknowledges support from the Research Foundation - Flanders ("Fonds voor Wetenschappelijk Onderzoek - Vlaanderen," G0D5117N). G.B. acknowledges support from the Internal Funds KU Leuven (Grant No. C14/18/094). G.B. and N.B. acknowledge support from the Research Foundation - Flanders ("Fonds voor Wetenschappelijk Onderzoek - Vlaanderen," G0E1420N, G098321N). The funders had no role in the study design, data analysis, data interpretation, and in the writing of this report.

## Author contributions

J.F., M.H., M.P., L.K., and T.K. designed the study and contributed to experiment design and data interpretation. M.H., J.F., S.L.H., N.B., S.C., S.K., and G.B. collected sequence and associated metadata. S.L.H., N.B. and G.B. performed phylogeographic analyses. J.F., L.K., M.H., S.K., S.C., and A.W. performed statistical analyses of patient metadata or analyzed next-generation sequencing data. A.P., A.L., M.Sa., N.N., J.K.O., R.R., A.B., C.A.K., A.O., E.S.L., V.E., and E.A.O. provided sequencing data of SARS-CoV-2. T.K., L.K., J.B., D.S., S.U., S.W., G.K., P.K., M.H.U., and L.J. performed experiments and analysed the data. J.F., L.K., T.K., P.K., S.L.H., and N.B. wrote the paper. J.F., G.D., and S.L.H. visualized the data. M.P., M.Sc., and G.K. were involved in funding acquisition.

## Funding

## Competing interests

The authors declare no competing interests.
