## [Peer Review File · Nature Communications]

Antibody escape and global spread of SARS-CoV-2 lineage A.27REVIEWERS' COMMENTS

Reviewer #1 (Remarks to the Author):

This manuscript details the A.27 SARS-CoV-2 lineage, globally this lineage is of little consequence with respect to raw numbers and has really only been observed in 500 or so cases, however the implications of what is present within A.27 are important and need to be published to highlight the convergent evolutionary potential of SARS-CoV-2. The spike protein mutations L18F, L452R and N501Y are the three key mutations present in A.27, with L452R also present in Delta, and N501Y present in Alpha, obviously the combination of these is alarming, and also interesting given an A lineage has independently acquired them both. Thankfully A.27 has now died out, seemingly due to dominance of Delta. The presented work is a thorough and detailed analysis of this curious lineage and its characteristics and should be published, its key finding is the propensity for lineages to independently acquire immune escape variants. I have spotted one typo of A27 on line 117 which should be corrected to A.27.

Reviewer #2 (Remarks to the Author, also please see attached):

Noteworthy results - The A.27 showed vaccine and COVID19 convalescent antibody resistance. Specific Monoclonal antibodies were not able to or have increased resistance to virus neutralization. A.27 does not contain the Spike D614G substitution, and it is attenuated in mouse models. The authors suggest it is due to a deletion in ORF3a. Finally, the authors support an earlier suggested origin of A.27 isolate to Western Africa. This study is of great interest to the community even if the virus did not become the dominant variant in the areas of study.

Additional studies suggested – It would be of great interest to know if the attenuation of A.27 is due to the lack of D614G or if it was due to the ORF3a deletion as suggested. Further information on lung titers and pathology would also be helpful.

Additional comments specific to the submission:

- Figure 1d can this be displayed at A.27 cases relative to other SARS-COV-2 cases.
- Page 33, Figure 4a – non-synonymous mutations relative to what genome – reference?
- Page 33, Figure 4c - Please bold or highlight the A.27 75% consensus substitutions that are found the VOC and VOI
- Page 33, Figure 4c. Please label the table with WHO nomenclature in addition to the lineage information to make it easier to follow for the reader. The AY.2 and AY.1 should be grouped with Delta.
- Page 34, figure 5a – what cell line was the virus passaged in for this figure? VeroE6 – please indicate this in the legend
- The subsequent figure (5 and 6) use a B.1 isolate for comparison. Why was this selected and what is the substitution of the this isolate relative to the A.27? Including this in figure 4c or figure 5a would be helpful.
- Page 22 line 659 – collected through June 1st? mentioned later but relevant here.
- Sampling bias for Bayesian analysis – Germany 50%? There are a total of 3907 isolates with 2545 from Africa nextstrain
- Page 26 lines 789-794, how were the viral stocks confirmed and originally propagated? I understand that the A.27 was confirmed by sequencing but were the other provided viral stocks sequenced? Please clarify. This could be helpful when looking at the growth rate of the B.1.351 isolate in Calu3 cells. Was the furin cleavage site present in this isolate? The growth of the virus in the Calu3 cell line is cell surface dependent and requires the presence of the furin cleavage sequence. Peacock, T. P., Goldhill, D. H., Zhou, J., Baillon, L., Frise, R., Swann, O. C., et al. (2021). The furin cleavage site in the SARS-CoV-2 spike protein is required for transmission in ferrets. *Nature Microbiology*, 1–19.
<http://doi.org/10.1038/s41564-021-00908-w>
- Page 4, Line 118 and page 12 line 344 - Please define where Mayotte is located. The “overseas region of France,” is not helpful.
- Page 4, line 124, The time period listed is confusing. Is it Dec 2020 through June 2021 or Dec 2020 through Aug, 2021. Please clarify.
- Extended figure 2 - A lighter coloring for the outgroup would be helpful to differentiate it from A.27.
- Page 5 and 6 – please place the tMRCA global phylogeny date in context of the introduction of the A.27 to Germany and France. Is the timeline biased by the oversampling in these countries?
- Page 7 line 197 - Please define the down selection criteria for the lineage defining mutations.
- Page 8 line 238 – has this growth defect been seen by others. No reference is provided.
- Page 9 242, remove “As” from the sentence.
- Page 9 line 247 and page 13 line 382 – Localization alteration would require co-staining with a cellular protein. The reference provided used LAMP1. This data would be helpful to confirm this localization.

Reviewer #1 (Remarks to the Author):

This manuscript details the A.27 SARS-CoV-2 lineage, globally this lineage is of little consequence with respect to raw numbers and has really only been observed in 500 or so cases, however the implications of what is present within A.27 are important and need to be published to highlight the convergent evolutionary potential of SARS-CoV-2. The spike protein mutations L18F, L452R and N501Y are the three key mutations present in A.27, with L452R also present in Delta, and N501Y present in Alpha, obviously the combination of these is alarming, and also interesting given an A lineage has independently acquired them both. Thankfully A.27 has now died out, seemingly due to dominance of Delta. The presented work is a thorough and detailed analysis of this curious lineage and its characteristics and should be published, its key finding is the propensity for lineages to independently acquire immune escape variants. I have spotted one typo of A27 on line 117 which should be corrected to A.27.

We thank the Reviewer for the evaluation of our manuscript and the kind feedback. We corrected A27 to A.27 (line 117).

Reviewer #2 (Remarks to the Author, also please see attached):

Noteworthy results - The A.27 showed vaccine and COVID19 convalescent antibody resistance. Specific Monoclonal antibodies were not able to or have increased resistance to virus neutralization. A.27 does not contain the Spike D614G substitution, and it is attenuated in mouse models. The authors suggest it is due to a deletion in ORF3a. Finally, the authors support an earlier suggested origin of A.27 isolate to Western Africa. This study is of great interest to the community even if the virus did not become the dominant variant in the areas of study.

Additional studies suggested – It would be of great interest to know if the attenuation of A.27 is due to the lack of D614G or if it was due to the ORF3a deletion as suggested. Further information on lung titers and pathology would also be helpful.

We also thank the second Reviewer for the evaluation of our manuscript and the thorough comments on the text and figures. The suggestion to investigate the attenuation of A.27 in vivo is intriguing and we indeed have plans to analyse the A.27 in vivo attenuation in more detail. However, we have so far not generated data regarding lung pathology or titers. Please find below our point-by-point responses to your comments.

Additional comments specific to the submission:

- 1.) Figure 1d can this be displayed at A.27 cases relative to other SARS-COV-2 cases.

The major concern is that we can only give percentages of A.27 to total sequenced samples available on GISAID in the respective countries in a particular timeframe, which does not necessarily reflect

the actual number of A.27 cases as some African countries sequenced only a few SARS-CoV-2 genomes. This likely over- or underestimates the actual A.27 cases and could be misinterpreted. Therefore, we would rather show the raw number of A.27 sequences as is.

2.) Page 33, Figure 4a – non-synonymous mutations relative to what genome – reference?

Thank you for this hint. We added the reference in the legend of Figure 4 (line 970).

3.) Page 33, Figure 4c - Please bold or highlight the A.27 75% consensus substitutions that are found the VOC and VOI.

We highlighted the 75 % consensus mutations in Figure 4c.

4.) Page 33, Figure 4c. Please label the table with WHO nomenclature in addition to the lineage information to make it easier to follow for the reader. The AY.2 and AY.1 should be grouped with Delta.

Thanks for the suggestion. We grouped AY.1/2 to Delta.

5.) Page 34, figure 5a – what cell line was the virus passaged in for this figure? VeroE6 – please indicate this in the legend.

The viruses were isolated and cultured on VeroE6 cells. We have now added this in the legend of Figure 5a (lines 986/987).

6.) The subsequent figure (5 and 6) use a B.1 isolate for comparison. Why was this selected and what is the substitution of the this isolate relative to the A.27? Including this in figure 4c or figure 5a would be helpful.

Thank you for addressing this ambiguity. The B.1 isolate came from the first German SARS-CoV-2 case in Munich early in the pandemic and is highly similar to Wuhan-Hu-1. The only non-synonymous substitution is D614G. As we did not initially clarify this, we now state this in the text with the name of the isolate (Muc-IMB-1) in lines 243/244. The GISAID accession number of the isolate is provided in the Methods section (line 579).

7.) Page 22 line 659 – collected through June 1st? mentioned later but relevant here.

We now added the collection period in the Data acquisition of the Methods (line 444/445).

8.) Sampling bias for Bayesian analysis – Germany 50%? There are a total of 3907 isolates with 2545 from Africa nextstrain.

We thank the Reviewer for this comment, as this was indeed not clearly described in the Results. For an initial estimation of the overall SARS-CoV-2 phylogeny containing the A.27 clade, we used the 2545 sequences from the Africa-focused nextstrain build, complemented with all available A.27 sequences (and removing duplicates). In a subsequent step, we estimated a more targeted phylogeny on the subtree containing the available A.27 sequences, complemented with 25 non-A.27 sequences (lines 144-146) which we found to yield sufficient temporal signal. Therefore, we performed our Bayesian phylogeographic analysis on this subtree. However, here sequences from Germany and France indeed biased the analysis. To mitigate this issue, we performed the described

subsampling. This is explained in the Methods section (lines 479 – 544). To appreciate this point, we now clarify which dataset was used for our BEAST analysis in the Results section (lines 151/152).

9.) Page 26 lines 789-794, how were the viral stocks confirmed and originally propagated? I understand that the A.27 was confirmed by sequencing but were the other provided viral stocks sequenced? Please clarify. This could be helpful when looking at the growth rate of the B.1.351 isolate in Calu3 cells. Was the furin cleavage site present in this isolate? The growth of the virus in the Calu3 cell line is cell surface dependent and requires the presence of the furin cleavage sequence. Peacock, T. P., Goldhill, D. H., Zhou, J., Baillon, L., Frise, R., Swann, O. C., et al. (2021). The furin cleavage site in the SARS-CoV-2 spike protein is required for transmission in ferrets. *Nature Microbiology*, 1–19. <http://doi.org/10.1038/s41564-021-00908-w>

The Reviewers concerns are well placed as we also observe cell culture adaptations during isolation or cultivation of SARS-CoV-2 on VeroE6. Therefore, we checked every isolated virus and the subsequently produced virus stocks using Illumina sequencing. Virus isolates with adaptations in the furin cleavage sites were not considered for further experiments. All virus stocks used in this study have identical consensus genomes to the GISAID accession numbers provided in the Material and Methods section. We re-checked the sequencing results for the B.1.351 stock used in the experiment and it is identical to the provided GISAID accession number. There was also no mutation in the furin cleavage site in variant frequencies above 10 % (our variant calling cut-off). Notably, this virus isolate has the curious S 242-244 deletion reported in a subset of B.1.351 genomes, which might have cause the observed phenotype. However, we do not have data supporting this hypothesis and therefore do not discuss this in the manuscript.

We now state in the Material and Methods sections that we checked our virus stocks using Illumina sequencing (lines 585-586).

We provide a reference to a publication that observed a similar phenotype in the growth of B.1.351 (see also our response to comment 15)

10) Page 4, Line 118 and page 12 line 344 - Please define where Mayotte is located. The “overseas region of France,” is not helpful.

We now define the location of Mayotte (line 118/119 and 352).

11) Page 4, line 124, The time period listed is confusing. Is it Dec 2020 through June 2021 or Dec 2020 through Aug, 2021. Please clarify.

We thank the reviewer, as this was an unintentional leftover from a previous version of the manuscript. We deleted August 24th from the sentence (line 125).

12) Extended figure 2 - A lighter coloring for the outgroup would be helpful to differentiate it from A.27.

We updated the colour for the outgroup to a lighter grey (Extended figure 2).

13) Page 5 and 6 – please place the tMRCA global phylogeny date in context of the introduction of the A.27 to Germany and France. Is the timeline biased by the oversampling in these countries?

We do not think that the timeline was biased due to the higher amounts of sequences in Germany and France, as we apply subsampling on the dataset to account for the sampling bias (see also the response to point 8 above). However, the small amount of early sequences results in an uncertainty of the tMRCA prediction ranging from mid-August 2020 to late October 2020 (see lines 156/157). We now put the A.27 introductions into Europe (Germany and France) in the context to its tMRCA (lines 164-166).

14) Page 7 line 197 - Please define the down selection criteria for the lineage defining mutations.

To the best of our knowledge, there is no widely accepted definition for lineage-defining mutations, as this can be calculated using different rationales e.g., sequence-based or phylogeny-based. We defined this in accordance with the outbreak.info database that defines lineage characteristic mutations as mutations present in at least 75 % of the lineage. We now state this in the Methods section (line 552). We also rephrased the sentence in line 197 (now line 202).

15) Page 8 line 238 – has this growth defect been seen by others. No reference is provided.

Thank you for this comment. We now provide a reference to a recent publication from December 2021 in which the authors observed similar growth differences for B.1.351 isolates in Calu-3 cells (<https://doi.org/10.3390/v14010023>) (line 246).

16) Page 9 242, remove “As” from the sentence.

We re-phrased and removed “As”. (lines 249/250)

17) Page 9 line 247 and page 13 line 382 – Localization alteration would require co-staining with a cellular protein. The reference provided used LAMP1. This data would be helpful to confirm this localization.

We agree with the Reviewer that such a staining would be beneficial and that our immunofluorescence analysis is not enough to claim that there are no differences in the subcellular localisation. We have therefore re-phrased this sentence (lines 390-393).